# A Temperature/pH Double-Responsive and Physical Double-Crosslinked Hydrogel Based on PLA and Histidine

**DOI:** 10.3390/gels8090570

**Published:** 2022-09-07

**Authors:** Qingrong Wu, Yu Fu, Wanying Yang, Shouxin Liu

**Affiliations:** Key Laboratory of Applied Surface and Colloid Chemistry, Ministry of Education, School of Chemistry and Chemical Engineering, Shaanxi Normal University, Xi’an 710119, China

**Keywords:** temperature/pH double sensitive, stereocomplex, hydrogen bond, metal coordination bond, physical double crosslinked

## Abstract

Hydrogel is a good drug carrier, widely used in the sustained-release aspect of tumor drugs, which can achieve the continuous release of drugs to the tumor sites. In this study, diethylene glycol monomethyl ether methacrylate (MEO_2_MA) and poly (ethylene glycol) methyl ether methacrylate (OEGMA) are temperature-sensitive monomers. N-Methacryloyl-L-Histidine (Mist) is pH sensitive monomer and ligand for metal coordination bond. The temperature-sensitive monomers and pH sensitive monomer with stereocomplex of modified polylactic acid (HEMA-PLLA_30_/PDLA_30_) were mixed, under 2,2’-azobis (2-methylpropionitrile) (AIBN) as radical initiator, polymer was formed by free-radical polymerization. The polymer was then immersed in ZnSO_4_ solution, the imidazole group of Mist monomer forms a tridentate metal coordination bond with Zn^2+^, temperature/pH double-responsive and physical double-crosslinked hydrogel was finally obtained. Comparing the hydrogen bond hydrogel, hydrogen bond and metal coordination bond double crosslinking hydrogel, metal coordination bond hydrogel, testing thermal stability, viscoelasticity, swelling, and morphology of three hydrogels. In addition, using UV-Visible spectroscopy (UV-Vis) to test the sustained release of the hydrophobic drug doxorubicin hydrochloride (DOX-HCl) in the human tumor environment (37 °C, pH = 5). We found that the temperature/pH double-responsive and physical double-crosslinked hydrogel had the most potential for the sustained drug release.

## 1. Introduction

Hydrogel is a high-molecular weight polymer with a three-dimensional network structure that has strong water absorption, but it is insoluble in water. Hydrogel is widely used in drug sustained release, tissue engineering, contact lenses, medical excipients, and cosmetic materials [1,2,3,4,5]. According to the cross-linking method, hydrogel can be divided into three types: physical cross-linking hydrogel [6,7,8,9,10], chemical cross-linking hydrogel [11,12], and physical-chemical double cross-linking hydrogel [13,14]. According to the type of stimulation response, they can be divided into temperature [15], pH [16], redox [17], optical [18], electric [19], magnetic [20], chemical [21] and other stimulation response hydrogel.

In medicine, when treating the disease, the drug release rate greatly affects the drug efficiency, and the hydrogel has three-dimensional network structure that can control the slow drug release. The tumor environment of the human body is different from the normal cell environment, in order to make the drug accurately play a role in the tumor site, the stimulus response function into hydrogel is a good choice. Temperature and pH are very important physiological parameters and easy to control, so temperature/pH dual responsive hydrogel is widely used in drug sustained release. Commonly used temperature-sensitive monomers are N-isopropyl acrylamide (NIPAM), N, N-diethylacrylamide, diethylene glycol monomethyl ether methacrylate (MEO_2_MA) and poly (ethylene glycol) methyl ether methacrylate (OEGMA). Commonly used pH-sensitive monomers are acrylic acid (AAC), methylacrylic acid (MAA), 2-(diethylamino) ethyl methacrylate (DEAEMA), 2-(Diethylamino) ethyl methacrylate (DMAEMA), etc. Gao et al. used NIPAM as temperature-sensitive monomer and AAC as pH-sensitive monomer, synthesized temperature/pH-sensitive hydrogel and performed drug slow release of acetylsalicylate acid, achieving drug encapsulation rate of 97.6% and cumulative release amount of 90.12% [22]. Zhou et al. used PDMAEMA as the temperature/pH double responsive polymer, synthesized the hydrogel with temperature and pH double responsiveness and studied the sustained-release behavior of the drug with 5-fluorouracil (5-Fu) as the model [23]. Ning et al. polymerized ε-caprolactone, MEO_2_MA, and OEGMA to form thermoresponsive micelles and performed sustained release of the hydrophobic drug, anethole [24]. In addition, from the perspective of the three gel cross-linking types, using chemical crosslinking agent can effectively enhance the mechanical properties of the gel, so people often use chemical cross-linking hydrogel and physical-chemical double cross-linking hydrogel to sustained release drugs, but chemical crosslinking agent is essentially toxic to the human body, so their biocompatibility is difficult to be satisfactory. To address this problem, physical cross-linked hydrogels are used to slow-release drugs, such as interionic electrostatic interactions, hydrogen bonds, metal coordination bonds, etc. Wang et al. introduced intermolecular hydrogen bonds generated by poly (N-acryloyl glycinamide) (PNAGA) to enhance the stability and mechanical properties of the hydrogel, assessing the gel release behavior of the drug (propranolol hydrochloride, PHCl) [25]. Wang et al. used Cellulose nanofibrils (CNFs) and cellulose nanocrystals (CNCs) for gel hydrogen bond force, using carboxymethyl chitosan (CMC) cationic group (amine groups) and sodium tripolyphosphate (TPP) anionic group between attraction provides ion interaction between the gel, thus had a variety of physical dynamic bond hydrogel and applied to drug delivery [26]. Tang et al. formed self-healing hydrogel formed by multivalent coordination of Ni^2+^ with polyhistidine (PHis) and multiple iminodiacetic acid (IDA) ligands, and studied the sustained release of rhodamine-modified PHis [27].

PLA is a hydrophobic high-molecular weight polymer, derived from renewable plants (such as corn), with non-toxic, good biocompatibility and biodegradable. The two enantiomers of PLA, PLLA and PDLA, are mixed in a 1:1 ratio, which can be stereocomplexed to form the physical force of hydrogen bonds, and can improve their thermal stability. The introduction of polylactic acid can provide the gel with hydrogen bonding [28,29]. L-histidine is an essential human amino acid, which is non-toxic, antibacterial and biocompatible. The imidazole group of histidine can combine with Zn^2+^ to form a tridentate metal coordination bond, and the introduction of histidine monomer can also have the physical force of coordination bond. Histidine also has pH sensitivity, where the carboxyl group, imidazole group and secondary amine group can be protonated or deprotonated, so the introduction of this monomer into the gel can confer gel pH sensitivity [30,31,32,33]. MEO_2_MA and OEGMA are temperature-sensitive monomers with hydrophilic properties and biocompatibility, and the two are mixed in different proportions with different LCST values, and introducing them can make the gel with temperature sensitivity [34,35].

In this study, we synthesized a temperature/pH double-responsive and physical double-crosslinked hydrogel, using AIBN as the radical initiator, polymerize the temperature-sensitive monomers MEO_2_MA:OEGMA = 90:10 (LCST = 37 °C), pH-sensitive monomer Mist and macromonomer HEMA-PLLA_30_:HEMA-PDLA_30_ = 1:1, and then coordinate the polymer with Zn^2+^ to obtain a physical double-crosslinked hydrogel. Structure of hydrogels were characterized, demonstrating the successful synthesis of macromonomer HEMA-PLLA_30_/PDLA_30_ and Mist by ^1^H NMR and FTIR, the successful formation of hydrogen bond by powder X-ray diffraction instrument (XRD), and the existence of metal coordination bond by X-ray energy dispersion analysis (EDAX). Next, testing performance of the hydrogen bond hydrogel, hydrogen bond and metal coordination bond double cross-linked hydrogel, and metal coordination bond hydrogel. Using thermal analysis system (TA) to test thermal stability of the gel, dynamic mechanics analyzer (DMA) to test viscoelasticity of the gel, the weighing method to test the swelling rate of the gel at different temperatures and pH conditions, environmental scanning electron microscope (ESEM) to observe three-dimensional network structure of the gel. Finally, the sustained release of the drug DOX-HCl in gel was tested using the UV-visible spectrophotometer (UV-Vis) in a simulated human tumor environment. The temperature/pH double-responsive and physical double-crosslinked hydrogel has great potential for sustained drug-release.

## 2. Results and Discussion

### 2.1. HEMA-PLLA_30_/PDLA_30_ and Mist Monomers

#### 2.1.1. HEMA-PLLA_30_/PDLA_30_ Macromonomers

HEMA-PDLA_30_ is formed by the polymerization of D-lactide (D-LA) with 2-hydroxyethyl methacrylate (HEMA). Appendix A shows the ^1^H NMR spectrum of HEMA-PDLA_30_, the peaks of 5.60 ppm (e) and 6.10 ppm (f) are the proton peaks of the carbon double bond, the peak of 1.93 ppm (b) is the methyl proton peak on the methyl propylene group, and the peak of 1.47–1.67 ppm (a) is the methyl proton peak of the D-LA repeat unit. Since the ^1^H NMR spectrum peaks of HEMA-PLLA_30_ and HEMA-PDLA_30_ are exactly the same, the repeat spectrum is no longer displayed. The above indicates that both the macromonomers HEMA-PLLA_30_ and HEMA-PDLA_30_ were successfully synthesized.

The formula for calculating the number of LA units in HEMA-PLLA_30_/PDLA_30_:(1)$$n=\frac{A_{d}}{A_{f}}+1,$$
where $A_{d}$ and $A_{f}$ stand for the integral area of peaks (*d*) and (*f*), respectively. Where *n* stands for the number of LA units in HEMA-PLLA_30_/PDLA_30_. $A_{f}$ = 1.00 and $A_{d}$ = 28.66, so *n* = 29.66 ≈ 30 [36]. 

Appendix A shows the FTIR spectrum of the HEMA-PDLA_30_. The peak of 1193 cm^−1^ is the expansion vibration peak of C–O–C, the peak of 1640 cm^−1^ is the expansion vibration peak of C=C, the peak of 1756 cm^−1^ is the expansion vibration peak of C=O, the peaks of 2945 cm^−1^ are the bending and expansion vibration peaks of –CH_3_ and –CH_2_, the peak of 3444 cm^−1^ is the expansion vibration peak of –OH. The above shows that the macromonomer HEMA-PDLA_30_ synthesis is successful. The FTIR spectrum of HEMA-PLLA_30_ and HEMA-PDLA_30_ are identical [37].

#### 2.1.2. Mist Monomer

The Mist monomer were obtained from a reaction of L-Histidine and methacryloyl chloride under the condition of NaOH. Appendix A shows the ^1^H NMR spectra of Mist monomer and L-Histidine monomer. Comparison of the peaks of Mist and L-Histidine shows the appearance of new signal peaks for Mist monomer. The peak of 1.69 ppm is the methyl proton peak on the methyl propylene group, the peaks of 5.21–5.47 ppm are the proton peaks of the carbon-carbon double bond. These demonstrating the successful synthesis of the Mist monomer [38].

### 2.2. Synthesis and Characterization of Temperature/pH Double-Responsive and Physical Double-Crosslinked Hydrogel

AIBN was the initiator, polymerizing MEO_2_MA, OEGMA, HEMA-PLLA_30_, HEMA-PDLA_30_, and Mist at 70 °C, and then the polymer was immersed in the ZnSO_4_ solution to obtain the temperature/pH double-responsive and physical double-crosslinked hydrogel. The composition and content of the gels are shown in Table 1. 

Appendix A shows the FTIR spectrum of the polymer which Mist grafted onto. The peaks of 2971 cm^−1^ and 1463 cm^−1^ are bending and expansion peaks of –CH_3_, the peak of 1723 cm^−1^ is the expansion vibration peak of C=O, the peak of 1125 cm^−1^ is the extended vibration peak of C–O–C, the peak of 625 cm^−1^ is the vibration peak of the imidazole group on the Mist monomer. Indicating that the Mist monomer is grafted onto the gel [39].

Figure 1 is X-ray diffractometer (XRD) spectra of the HEMA-PLLA_30_ macromonomer, HEMA-PDLA_30_ macromonomer and hydrogen-bonded hydrogel. Comparing the diffraction peaks of the three, new diffraction angles of 2 θ = 12°, 21° and 24° represent the stereo complexation diffraction peaks of HEMA-PLLA_30_ and HEMA-PDLA_30_, which proves the successful stereocomplex of HEMA-PLLA_30_ and HEMA-PDLA_30_. That is, the hydrogen bonds formed successfully [40].

Appendix A shows the EDAX of double-crosslinked hydrogel, the peak of 1.02 keV represents the energy spectrum peak of Zn elements. The monomers of synthetic gel are mostly macromolecular long chain, which contains a large number of C and O element, so the energy spectrum peak of N element and Zn element is relatively not obvious. However, even if the gel is soaked in double-distilled water for 3 days, free zinc is still detected in the gel, thus indicating that Mist and Zn^2+^ successfully formed the metal coordination bond [39].

### 2.3. Temperature Sensitivity of the Hydrogels

To investigate the temperature sensitivity of the hydrogels, the swelling and deswelling of hydrogels at 25 °C and 37 °C were tested. Figure 2A shows the swelling curves of the hydrogels at 25 °C, pH = 7.4, the swelling ratio of gel 2 and gel 3 is greater than that of gel 1, because both gel 2 and gel 3 contain hydrophilic monomer Mist which can increase the swelling ratio of the hydrogels. The swelling ratio of gel 2 is less than that of gel 3, because gel 2 contains hydrophobic macromonomers HEMA-PLLA_30_ and HEMA-PDLA_30_, and the two are stereocomplexed to form hydrogen bond. Hydrophobicity and hydrogen bond make the internal structure of gel 2 tighter, so the swelling ratio of gel 2 is less than the swelling ratio of gel 3.

Figure 2B shows the deswelling curves of the hydrogels at 37 °C and pH = 7.4, which shows that the hydrogels tend to shrink and lose water, which proves that the hydrogels have temperature sensitivity. At 25 °C, the hydrophilic polymer chain P (MEO_2_MA-co-OEGMA) form the hydrogen bond with water, increasing the swelling ratio of hydrogels. While at 37 °C, the hydrogen bond between the hydrophilic polymer chain P (MEO_2_MA-co-OEGMA) and water are partially destroyed, resulting in the water loss of the hydrogels, and thus in a state of deswelling.

### 2.4. The pH Sensitivity of the Hydrogel

To study pH sensitivity of the hydrogels, the swelling and deswelling of hydrogels at pH = 5 and pH = 7.4 were tested. Figure 3A shows the swelling curves of the hydrogels at 37 °C, pH = 5. The swelling ratio of gel 2 and gel 3 is greater than that of gel 1, because gel 2 and gel 3 contain the pH sensitive monomer Mist. The Mist monomer contains three groups that can obtain or lose protons, namely, the carboxyl group, imidazole group, and secondary amine group. When pH = 5, the carboxyl group is negatively charged, the imidazole group is positively charged, and the secondary amine group is positively charged, so the Mist monomer is overall positively charged. The positive electricity will enhance the repulsion of Mist monomers, thus increasing the spacing of Mist monomers, so increase the pore size of the hydrogels, and correspondingly increase the swelling ratio of the hydrogels. Molecular structural formulas of the Mist monomer under different pH were shown in Figure 1.

Figure 3B shows the deswelling curves of the hydrogels at 37 °C, pH = 7.4, which shows that the swelling rate is constantly decreasing, which proves that the hydrogels have pH sensitivity. When pH = 7.4, the carboxyl group is negatively charged, the imidazole group is not charged, and the secondary amine group is positively charged, so the Mist monomer is overall electroneutral. The electroneutrality does not widen the spacing between Mist monomers, so the swelling ratio of the hydrogels at pH = 7.4 is less than that of pH = 5. Molecular structural formulas of the Mist monomer under different pH were shown in Figure 1.

### 2.5. Mechanical Properties of the Hydrogel

To study the viscoelasticity of the gel, the energy storage modulus *E’* and the loss tangent tan *δ* were tested with the shape variable of 8%. The *E’* can reflect the elasticity of gels, the larger the *E’*, the greater the elasticity of gels. Figure 4A shows curves of *E’* of the gels with frequency, *E’* increases with frequency. At 10 Hz, the *E′* of gel 1, gel 2, and gel 3 is 477 KPa, 126 KPa, 50 KPa, respectively, and the *E′* of gel 2 is lower than the *E′* of gel 1. Although the hydrogen bond and metal coordination bond can enhance its elasticity, the pH sensitivity can increase the swelling ratio of the gels and reduce its elasticity. The *E’* of gel 2 is greater than the *E’* of gel 3, because the hydrogen bond force between HEMA-PLLA_30_ and HEMA-PDLA_30_ and their own hydrophobicity make the gel 2 pore size smaller. The smaller the pore size, the larger the mechanical properties, so the *E’* of gel 2 is larger is greater than the *E’* of gel 3.

The loss tangent tan *δ* can reflect the viscoelasticity of the gels, the larger the tan *δ*, the greater the viscosity, the smaller the tan *δ*, the greater the elasticity. Figure 4B is curves of the tan *δ* of the gels with the frequency, and the tan *δ* of the gels increase with the frequency. At any same frequency, tan *δ* of gel 3 is greater than gel 2, because coordination bond is weaker than the combination of hydrogen bond and coordination bond, gel 3 has more viscosities.

### 2.6. Thermogravimetric Analysis (TGA) of Hydrogels

The thermal stability was compared by testing the thermal weight of hydrogen bond hydrogel, double crosslinked hydrogel, and metal coordination bond hydrogel. Figure 5 shows TGA curves of the three hydrogels, the first weight loss of the gel is the thermal degradation temperature, so the thermal degradation temperature of the gel 1, gel 2, gel 3, respectively, about 220 °C, 190 °C and 190 °C, indicating that the thermal stability of gel 1 is the best. Which is mainly because the stereocomplex of HEMA-PLLA_30_ and HEMA-PDLA_30_ enhances the thermal stability of the gel 1. The thermal stability of gel 2 and gel 3 is about the same, because the addition of hydrophilic monomer Mist weakens the internal structure of the gel and weakens the thermal stability of the gel.

### 2.7. Environmental Scanning Electron Microscope (ESEM) Analysis of Hydrogels

The ESEM images of the gels can visually show the three-dimensional network structure of the gels, comparing the pore size relationship of different gels at the same magnification. Figure 6 is ESEM map of gel 1, gel 2, and gel 3 immersed in double-distilled water at 25 °C, pH = 6. Which shows that the pore size of gel 1 is small to unobserved, because the hydrophobic monomers HEMA-PLLA_30_ and HEMA-PDLA_30_ in gel 1 make the gel less hydrophilic, so the interior is tighter. The pore size of gel 2 is smaller than that of gel 3, because it has hydrophobic monomers HEMA-PLLA_30_ and HEMA-PDLA_30_, also has more physical cross-linking points of hydrogen bond, so the pore size of gel 2 is smaller than that of gel 3.

### 2.8. Drug Release of the Hydrogel

The drug load capacity of gel 1, gel 2, and gel 3 were 3.1%, 8.6%, and 10.2%, respectively, which is positively correlated with the pore size of the gels. The pore size of the gels is gel 3 > gel 2 > gel 1, so the drug load capacity of the gels is also gel 3 > gel 2 > gel l. Figure 7 shows the slow-release curves of the load DOX-HCl gels under the condition of the simulated human tumor environment of 37 °C, pH = 5. It can be seen that the gel release rate is fast in the first 12 h and slow in the last 12–80 h. The final amount of drug release of gel 1, gel 2 and gel 3 are 16%, 42% and 55%, respectively, because pore size of gel 1 is the smallest, pore size of gel 2 is medium, the pore size of gel 3 is the largest, the larger the pore size, the greater the amount of drug release. Although the amount of drug release of gel 3 is greater than that of gel 2, the mechanical properties of gel 3 are very poor and easy to break, while gel 2 has good mechanical properties and uniform pore size. Considering many factors, gel 2 is more suitable for sustained release drugs. The slow release of DOX-HCl improves the situation of adverse reactions due to the drug release of too fast and excessive, so the gel has a wide application prospect in the field of sustained drug-release.

## 3. Conclusions

In this study, we formed a double-crosslinked hydrogel with the hydrogen bond through the stereocomplex of the macromonomers and the metal coordination of the imidazole group of Mist with Zn^2+^. Temperature-sensitive and pH-sensitive monomers polymerize via free radicals, giving the gel temperature with pH sensitivity. Comparing hydrogen bond hydrogel, double-crosslinked hydrogel and metal coordination bond hydrogel, from the aspect of mechanical properties and thermal stability, hydrogen-bond hydrogels have the strongest mechanical properties and thermal stability. From the pore size, the pore size of the double-crosslinked hydrogel is the most uniform. From the drug sustained release, the metal coordination bond hydrogel has the highest amount of drug release. However, the drug sustained release hydrogel should have good mechanical properties, appropriate drug release rate, and high amount of drug release, combined on these requirements, the conclusion is that double-crosslinked hydrogel is a good carrier of drug sustained release. The drug load capacity of double-crosslinked hydrogel was as high as 8.6%, and the amount of drug sustained release was as high as 42%, so it has a good application prospect in the field of drug slow release.

## 4. Materials and Methods

### 4.1. Materials

2-Hydroxyethyl methacrylate (HEMA, 99%) was purchased from J&k (Beijing, China). L-lactide (L-LA, 98%) and poly (ethylene glycol) methyl ether methacrylate (OEGMA, 95%, Mn = 475 g·mol^−1^) were purchased from Aladdin (Shanghai, China). D-lactide (D-LA, 99%) was purchased from Damas-beta (Shanghai, China). 1,8-Diazabicyclo [5.4.0] undec-7-ene (DBU, 99%) was purchased from Rhawn (Shanghai, China). L-Histidine (L-His, 99%), methacryloyl chloride (95%), diethylene glycol monomethyl ether methacrylate (MEO_2_MA, 97%, Mn = 188.22 g·mol^−1^), 2,2’-azobis (2-methylpropionitrile) (AIBN, 98%) were purchased from Macklin (Shanghai, China). ZnSO_4_ 7H_2_O (Mn = 287.56 g·mol^−1^) was purchased from Jinhuada (Guangzhou, China). CH_2_Cl_2_ (DCM) was dried over CaH_2_, THF was super dry solvent purchased from Energy Chemical (Anqing, China). All other chemicals were analytical grade and were used without further purification.

### 4.2. Synthesis

#### 4.2.1. Synthesis of HEMA-PLLA_30_ and HEMA-PDLA_30_ Macromonomers

L-LA (1 g, 6.9 mmol) was added to a 50 mL three-neck flask, 20 mL DCM was added as a solvent, the N_2_ was then passed into the three-neck flask for 5 min, add the HEMA again (56 μL, 0.46 mmol) and DBU (100 μL, 0.65 mmol). After 12 h at room temperature of the reaction, the reaction was terminated by addition of the excess benzoic acid. Half an hour later, the reaction mixture was added to cyclohexane under ice bath condition. After precipitating the product, pour out the supernatant, dry the product in a fume hood, the final white solid is macromonomer HEMA-PLLA_30_. Similarly, macromonomer HEMA-PDLA_30_ shares the same procedure described above. The synthetic route was shown in Figure 2A.

#### 4.2.2. Synthesis of Mist Monomer

Mist was prepared according to the method in Okamoto [41]. L-Histidine (5 g, 0.032 mol) was added to a 50 mL three-neck flask and dissolved in NaOH (2.52 g, 20 mL) solution. In the ice bath environment, the system mixed the methacryloyl chloride (3.65 mL, 0.038 mol) with the 1,4-dioxane of 10 mL and slowly dropped into the system. After drip addition, the ice bath environment was removed and allowed to react at room temperature for 1 h. Then the reaction mixture was distilled under reduced pressure at 65 °C to remove the 1,4-dioxane. Then use HCl solution to adjust pH for 2, using ether extraction three times to remove unreacted raw materials and by-products, collect water layer solution. Then use NaOH solution to adjust pH for 5, with ethanol extraction products, the ethanol solution was distilled under reduced pressure to remove a large amount of ethanol. The concentrated solution was washed with acetone to obtain a white solid, dissolved in a moderate amount of water and freeze-dried, the white powdered Mist monomer was finally obtained. The synthetic route was shown in Figure 2B.

#### 4.2.3. Synthesis of Hydrogen-Bonded Hydrogel

In 10 mL flask, mixing HEMA-PLLA_30_ (0.1 g, 0.043 mmol) and HEMA-PDLA_30_ (0.1 g, 0.043 mmol) and dissolving them with 4 mL of the superdry solvent THF, seal the bottleneck. The 2 h ultrasound enables HEMA-PLLA_30_ and HEMA-PDLA_30_ to form hydrogen bond. The synthetic route of stereocomplex was shown in Figure 2C. Then reduced pressure distillation removed THF, add the MEO_2_MA (830 μL, 4.7 mmol) and OEGMA (104 μL, 0.252 mmol). Stir and dissolve the stereocomplex of HEMA-PLLA_30_ and HEMA-PDLA_30_, add to AIBN (0.008 g, 0.049 mmol), closed and filled with N_2_, the reaction was stirred in 70 °C oil bath for 1 h, hydrogen-boned hydrogel was obtained.

#### 4.2.4. Synthesis of a Coordination-Bond Hydrogel

In a 10 mL flask, add Mist (0.4 g, 1.79 mmol), 1 mL of double-distilled water, stir and dissolve, then add the MEO_2_MA (830 μL, 4.7 mmol), OEGMA (104 μL, 0.252 mmol) and AIBN (0.008 g, 0.049 mmol). Stir and mix well and seal it, fill it with N_2_, stir the reaction in 70 °C oil bath for 3 h, to obtain the polymer. After deprotonating the imidazole group of the Mist monomer with a KOH solution of pH 12, and then soaking in a 0.1 M ZnSO_4_ solution, the zinc ion formed a tridentate metal coordination bond with the imidazole group. The hydrogel with metal coordination bond was obtained.

#### 4.2.5. Synthesis of Double-Crosslinked Hydrogel

HEMA-PLLA_30_ (0.1 g, 0.043 mmol) and HEMA-PDLA_30_ (0.1 g, 0.043 mmol) were added to a 10 mL flask, dissolved with 4 mL of superdry solvent THF and sealed it, ultrasonic for 2 h to fully form hydrogen bond. Reduced pressure distillation to remove THF, MEO_2_MA (830 μL, 4.7 mmol) and OEGMA (104 μL, 0.252 mmol) were added to dissolve the stereocomplex of HEMA-PLLA_30_ and HEMA-PDLA_30_. Mist (0.4 g, 1.79 mmol) was dissolved in 1 mL of double-distilled water, then added the Mist soluton to the flask. Then add AIBN (0.008 g, 0.049 mmol) and seal the bottleneck, fill in N_2_ to flask, and the reaction was stirred in a 70 °C oil bath for 3 h to obtain the gel polymer. The polymer was soaked in the KOH solution of pH 12 for 24 h to deprotonate the imidazole group of the Mist monomer. Then the polymer was transferred to 0.1 M ZnSO_4_ solution to form tridentate metal coordination bond of the zinc ion with the imidazole group, and finally obtained a double-crosslinked hydrogel with hydrogen bond and coordination bond. The synthetic route was shown in Figure 2D.

### 4.3. Methods

#### 4.3.1. Spectral Characterization

The FTIR was tested using a type Frontier Mid-Far-Infrared spectrometer produced by PerkinElmer. The ^1^H-NMR was tested using a type ECZ400S/L NMR spectrometer produced by Japan Electronics Corporation, Inc, test samples were performed with D_2_O as the solvent. The XRD was tested with a type D8 Advance powder X-ray diffractor produced by Bruker at 25 °C and operated at a voltage of 40 kV and current of 40 mA, with a scan rate of 2°/min and a scan range of 5° to 40°. The EDAX was tested using the energy spectrum analysis function of the type Quanta 200 environmental scanning electron microscope manufactured by FEI in the Netherlands, hydrogel was frozen with liquid nitrogen, then dried in a freeze-dryer, and sprayed with gold with ion sputter for 30 s before testing.

#### 4.3.2. Thermogravimetric Analysis of the Hydrogels

The thermal stability of the frozen-dried hydrogel was studied at 5 mg by using the thermal analysis system of the type Q-600 manufactured by TA under the conditions of N_2_ atmosphere, 10 °C/min velocity, and a temperature range of 25 °C to 600 °C.

#### 4.3.3. ESEM Analysis of the Hydrogels

The swelling-balanced gel was frozen in liquid nitrogen, dried with a freeze-dryer, and then the dried gel was sprayed with gold for 60 s with ion sputter, the morphology of the gel was visualized using a type Quanta 200 environmental scanning electron microscope produced by FEI in the Netherlands. 

#### 4.3.4. Mechanical Properties Analysis of the Hydrogels

The hydrogel was cut into neat cuboid blocks and tested using the 242 E dynamic mechanical analyzer (DMA) produced by Germany Chi Instrument Manufacturing Co., Ltd. In standard mode, the test temperature was 37 °C, and the E’ and tan δ were measured to compare the mechanical strength of the three gels.

#### 4.3.5. Swelling−Deswelling Behavior of the Hydrogels

The swelling property of the gels was tested by the weighing method. The freeze-dried hydrogel was weighed and then immersed in double-distilled water under different conditions (25 °C, pH = 7.4; 37 °C, pH = 5). The gel was removed at fixed time intervals, drained the surface water with filter paper and weighed with electronic balance. The swelling rate of the gels is calculated as follows:(2)$$\mathrm{Swelling}\mathrm{ratio}=(W_{t}-W_{d})/W_{d},$$
where $W_{t}$ is the swelling gel mass removed at the fixed time interval, and $W_{d}$ is the mass of the dry gel before swelling.

Similarly, the deswelling ability of gels was tested using the weighing method. The gel described above drained surface water and weighed, and then placed in double-distilled water at 37 °C, pH = 7.4. The above method was used to weigh the fixed time intervals, the deswelling rate of the gels is calculated as follows:(3)$$\mathrm{Deswelling}\mathrm{ratio}=(W_{t}-W_{d})/W_{s},$$
$W_{t}$ is the gel mass removed at the fixed time interval, $W_{d}$ is the mass of the dry gel before swelling, and $W_{s}$ is the mass of the water absorbed by the equilibrium gel before deswelling.

#### 4.3.6. Drug Release of the Hydrogels

Preparation of load DOX-HCl gel: Weigh 50 mg of freeze-dried gel, placed in 10 mg/mL of DOX-HCl solution (50 mg DOX-HCl, 5 mL THF). Swelled at room temperature for 3 days, wash the gel surface with THF and then placed in a 70 °C vacuum drying oven for 1 day for the later slow-release DOX-HCl test.

Determination of the maximum absorption wavelength of DOX-HCl: using PBS buffer with pH = 7.4 as the blank control, the absorbance of DOX-HCl solution (The PBS buffer with pH = 7.4 was used as a solvent) at 200 nm to 600 nm was tested by a UV-Vis spectrophotometer, knowing that the maximum absorption wavelength was 233.5 nm.

Determination of the standard curves of DOX-HCl solution: Using a PBS buffer of pH = 7.4 as the solvent, DOX-HCl solution was prepared at 1 μg/mL, 2 μg/mL, 5 μg/mL, 10 μg/mL, 15 μg/mL, 20 μg/mL, 25 μg/mL, 50 g/mL. Absorbance was tested with a UV-Vis spectrophotometer at a maximum absorption wavelength of 233.5 nm. With the absorbance as the ordinate and the solution concentration as the abscissa, the linear relationship of the Abs and c as follows:*Abs* = 0.05687*c* + 0.04795, *R*^2^ = 0.99891,(4)
where *Abs* is the absorbance of the DOX-HCl solution at 233.5 nm, and *c* is the concentration of the DOX-HCl solution.

Test of gels sustained release amount of DOX-HCl: The above dried carrier gel was soaked in 150 mL pH = 7.4 PBS buffer, and 3 mL of buffer was removed at a fixed time interval to test its absorbance, and then 3 mL of fresh buffer was added to the drug slow-release system. The test was not stopped until the gel no longer continued with the sustained-release drug. According to the standard curve equation of DOX-HCl solution, the concentration of the solution is derived by the absorbance, and thus the cumulative release of the drug is calculated, the calculation formula as follows:(5)$$\mathrm{Cumulative}\mathrm{release}(\%)=\frac{V_{e}\sum_{1}^{n-1}C_{i}+V_{0}C_{n}}{m_{drug}}\times 100,$$
where $V_{e}$ represents the volume of solution extraction each time, namely 3 mL; $c_{i}$ represents the concentration of each solution extraction, *n* represents the total number of times of solution extraction, and $V_{0}$ represents the volume of solution in the system after drug slow release, namely 150 mL.

Determination of the drug load capacity of gel: Weigh 50 mg of freeze-dried gel, loaded with DOX-HCl, and then weigh the quantity of the dried gel loaded with DOX-HCl. The formula for calculating the drug load capacity of gel as follows:(6)$$\mathrm{LC}\;(\%)=\frac{m_{gel}^{DOX-HCl}-m_{gel}}{m_{gel}}\times 100,$$
where *LC%* is the drug load capacity of gel, $m_{gel}^{DOX-HCl}$ is quantity of the dried gel loaded with DOX-HCl, $m_{gel}$ is quantity of the dried gel unloaded with DOX-HCl.

## Data Availability

Not applicable.

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
