# Peer review of "A Temperature/pH Double-Responsive and Physical Double-Crosslinked Hydrogel Based on PLA and Histidine"

_gels, 2022, doi:10.3390/gels8090570_

Round 1
Reviewer 1 Report
Reviewer Report
A Temperature / pH Double-Responsive and Physical Double-Crosslinked Hydrogel Based on PLA and Histidine
Summary:
This paper describes the synthesis and characterization of hydrogel networks based on HEMA-PLLA/HEMA-PDLA, ethylene glycol methacrylates and a histidine component. These gels combine the advantages of covalent, H-bond, and metal coordination crosslinking. Mechanical properties are characterized; temperature and pH sensitivity are demonstrated by a number of techniques, and reasonable (in most cases) explanations given for the observations. Release of DOX-HCL, as a model drug, was studied under simulated tumor conditions of temperature and pH. It is concluded that the best candidate for drug delivery applications is the hydrogel containing both hydrogen bonding/complexation and metal coordination bonding, as it combines an appropriate release rate with good mechanical properties.
Comments:
Language and writing
I strongly recommend the authors have the manuscript checked by a native English speaker to improve the grammar, tenses, punctuation etc. In many places it is hard to extract the intended meaning from the text.
Eg. for the sentence fragment ‘synthesized the temperature with the pH double response stimulation hydrogel’ (p2 line 53), it appears they mean that a hydrogel with temperature and pH double responsiveness was synthesized, but it is not immediately clear.
Another example: ‘The PKa of Mist monomer is about 6.5, which is less than PKa at pH=5, Mist protonated is positively charged’ (p6 line 205). The meaning is probably that pH 5 is lower than the pKa of Mist, therefore Mist is protonated at this pH. Rewording would make the meaning much clearer. Note also that PKa should be written with a lower case ‘p’ (pka).
There are quite a few instances of incorrect word choice or missing words.
Word choice: Eg. p8 line 226, ‘The loss factor tanδ can react the viscoelasticity of the gel’; p13 line 401, ‘secondary distilled water’ should probably be ‘double-distilled water’.
Missing words: eg. p2 line 74, ‘PLA is a hydrophobic high-molecular [weight] polymer’.
The abstract consists of only three sentences, including an 89-word sentence and a 77-word sentence with many commas.
There are far too many English language errors for me to individually address here. Please use a scientific proofreading service.
Scientific content
The explanations given for the observed results seem generally scientifically sound, however, lack detail in some cases. The study contributes some useful information to the field of material design for drug delivery.
MEO2MA is not a standard abbreviation and in fact doesn’t correctly convey the chemical structure. Diethylene glycol monomethyl ether is usually abbreviated as diEGME, DEGME, DGME or DEGMME.
OEGMA is referred to variably as ‘poly(ethylene glycol) methyl ether methacrylate’ (abstract and introduction) and ‘oligo (ethylene oxide) methacrylate’ (Materials and Methods section). Please keep the terminology consistent.
P3 line 109, PDLA should be DLA. Similarly, in Scheme 1 (A), the label PDLA should be DLA.
P3 lines 125-126 should be ‘The FTIR spectra of HEMA-PLLA30 and HEMA-PDLA30 are identical.’
The NMR spectrum in Figure 1 should be integrated to demonstrate the number of LA units incorporated. The structure is drawn as having 30 units, but no evidence for this is given. The feed ratio of monomers is not sufficient evidence for the ratio obtained in the final product.
P5 line 181-182, please expand on the following explanation: ‘Mist contains pH sensitive group carboxyl and imidazole group, negatively charged -COO- and positive charged imidazole group can produce attraction and repulsion, thus increasing the swelling rate’.
Attraction and repulsion of what and how does this influence swelling?
P6 line 200, at pH 5, both the COOH groups and imidazole groups would be largely protonated, however, the authors state it as -COO- (deprotonated). Please expand on the explanation of the type of interaction here and how this increases the pore size.
P7/8 line 223 and 224 (and numerous other places). What is meant by ‘aperture’ here? Do the authors mean ‘pore size’?
P8 lines 229 and 234, the terms ‘single coordination bond’ and ‘single hydrogen bond’ are used misleadingly. I think the authors mean these as the single method of crosslinking used in each case, rather than referring to a single bond of each type. This should be clarified.
P8 line 236/245, TA should be TGA. That is, ‘thermogravimetric analysis’ rather than ‘thermal analysis’, which is a more general term. On p13 line 387, TA is used to denote the manufacturer of the instrument, and this should here also be distinguished from the name of the technique itself (TGA).
Figure 12. It is very hard to visually appreciate from these images the size differences mentioned in the accompanying text. I suggest leaving this out altogether because these effects have been demonstrated more convincingly by the other techniques.
Section 2.8 Drug release of the hydrogel. What was the drug loading efficiency for each hydrogel? Was this characterized? Gel 3, having the largest pore size, could have taken up a greater quantity of drug, and likewise, gel 2 a greater quantity than gel 1. It is important because the greater apparent release rate by gel 3 might reflect both/either/or a greater loading vs higher intrinsic release rate. These two factors should be distinguished.
P10 lines 296 and 297, the meaning of the following is unclear: ‘But the drug slow release hydrogel should have good mechanical properties, appropriate drug slow release rate, and high drug release rate’. How can it have both a high release rate and a slow release rate?
The heading for section ‘4.2.1. Synthesis of HEMA-PDLA30 Macromonomer’ should be ‘4.2.1. Synthesis of HEMA-PDLA30 and HEMA-PLLA30 Macromonomers’, because both L and D forms were synthesized.
Reviewer 2 Report
In general, the main concept of the paper is interesting and worth investigating. However, the paper requires improvement to be acceptable for publication. Authors should pay attention to the following aspects:
1) Introduction of the paper should be corrected. For example, its first paragraph (i.e. 8 lines) is based on 21 references. Such notations as [15-21] constitute too much of a simplification and Authors should describe in few sentences given examples.
2) The paper contains numerous abbreviations thus an additional subsection should be added with all of them and their explanations.
3) Discussion over the results of the experiments should be compared with the results of other works and supported by references to these works. For example, there are no references in the analysis of obtained FT-IR spectra.
4) The language of the paper should be significantly improved. Additionally, the article contains many too long sentences (e.g. lines 167-173 or 178-184 are one sentence) thus the paper needs to be corrected stylistically, too.
5) Captions of Figures 7-8 should start with a capital letter. Additionally, y axes in these figures have no units.
6) Final Conclusions should be given in a more quantified manner.
7) Section 4.2.1.: in what temperature the synthesis of HEMA-PDLA30 Macromonomer was performed? This information should be added.
8) There is no information concerning the statistical analysis of obtained results. In how many replications the experiments were conducted?
9) Section References should be improved to be consistent and in line with the requirements of the Journal. For example, some references contain the whole journal names, and some contain their abbreviations - this needs to be unified.
Reviewer 3 Report
Comments for gels-1877244
The paper by Shouxin Liu and coworkers describes “A Temperature/pH Double-Responsive and Physical Double Crosslinked Hydrogel Based on PLA and Histidine”. The authors presented the details in the manuscript in a good format. Nevertheless, there are still some issues needed to be addressed before publication in this journal.
Comments:
1. In Figure 1, 1H NMR spectra of macromonomer HEMA-PDLA30 after 7.25 ppm, some impurities are visible, the authors need to provide purified NMR.
2. Authors need to provide the SEC study of polymers.
3. Authors must be provided the rheology study of hydrogels
4. Please add the specific experimental process for the calculation of the weight of the loaded drug.
5. Please add the cell viability and cytotoxicity experimental work of hydrogels, if the author provides basic biocompatibility studies of hydrogels it would be more impact on the manuscript.
6. Please check for grammar corrections.
Round 2
Reviewer 1 Report
Comments on revised manuscript gels-1877244
Most of the corrections have been made to my satisfaction. I’d like to note the following:
P3 line 121. Ad and Af should be the other way around, ie. Af = 1.00 and Ad = 28.66.
P9 line 274. The number ‘3’ for gel 3 is missing.
P9 line 279. It should be ‘drug release’ not ‘drug release rate’ because here you are not referring to rate but rather an amount released.
Abbreviations given in Table 2 are D-LA and L-LA but written as DLA on P3 lines 111 and 115.
In the added green text P14 lines 451 to 455 ‘quality’ should be ‘quantity’.
Comment 6. I know the authors are referring to the methacrylate, but I was suggesting something like DGMEMA instead of MEO2MA.
P8 line 241. I was referring to the word ‘react’ as being incorrect.
The English language is still not good, although some small attempt to improve it does seem to have been made.
Reviewer 2 Report
Paper has been mostly corrected in line with the recommendations thus the manuscript in its current form may be accepted for publication.
Author Response
Thank you very much for your review and evaluation of my manuscript, your review opinions have greatly improved my article.
Reviewer 3 Report
It can be accepted in the present form.
Author Response

(The authors gave the same response as above.)
